# Optimization of Radiolabeling of a [^90^Y]Y-Anti-CD66-Antibody for Radioimmunotherapy before Allogeneic Hematopoietic Cell Transplantation

**DOI:** 10.3390/cancers15143660

**Published:** 2023-07-18

**Authors:** Gordon Winter, Carmen Hamp-Goldstein, Gabriel Fischer, Peter Kletting, Gerhard Glatting, Christoph Solbach, Hendrik Herrmann, Elisa Sala, Michaela Feuring, Hartmut Döhner, Ambros J. Beer, Donald Bunjes, Vikas Prasad

**Affiliations:** 1Department of Nuclear Medicine, Ulm University Medical Center, 89081 Ulm, Germany; carmen.hamp-goldstein@uniklinik-ulm.de (C.H.-G.); fischer@euro-pet.de (G.F.); gerhard.glatting@uniklinik-ulm.de (G.G.); christoph.solbach@uniklinik-ulm.de (C.S.); hendrik.herrmann@uniklinik-ulm.de (H.H.); ambros.beer@uniklinik-ulm.de (A.J.B.); pvikas@wustl.edu (V.P.); 2Department of Internal Medicine III, Ulm University Medical Center, 89081 Ulm, Germany; elisa.sala@uniklinik-ulm.de (E.S.); michaela.feuring@uniklinik-ulm.de (M.F.); hartmut.doehner@uniklinik-ulm.de (H.D.); donald.bunjes@uniklinik-ulm.de (D.B.); 3Mallinckrodt Institute of Radiology, Division of Nuclear Medicine, Washington University in St. Louis, St. Louis, MO 63130, USA

**Keywords:** anti-CD66 antibody, *p*-SCN-Bn-CHX-A″-DTPA, yttrium-90, radioimmunotherapy, myeloablation, hematopoietic cell transplantation (HCT)

## Abstract

**Simple Summary:**

For high-risk patients, particularly young or elderly patients, with diseases of the hematopoietic cells, allogeneic hematopoietic cell transplantation (HCT) is often the only potentially curative treatment option. In preparation, high-dose chemotherapy and, in selected cases, external beam radiotherapy (EBRT) are often used. Especially in high-risk patients, the increased toxicity profile of these forms of treatment can mean treatment failure and even death. With high-dose radioimmunotherapy, the bone marrow can be treated with significantly less stress on the body. For a promising therapy targeting CD66, good results in previous treatments were demonstrated, but there has been increased non-specific accumulation in the liver and kidneys. The objective of this work was to use a recent chelator variant to improve radiolabeling and reduce non-specific accumulation with unaltered bone marrow uptake. In this article, we describe the production and demonstrate the improved distribution of the radiolabeled antibody based on early promising patient data.

**Abstract:**

For patients with acute myeloid leukemia, myelodysplastic syndrome, or acute lymphoblastic leukemia, allogeneic hematopoietic cell transplantation (HCT) is a potentially curative treatment. In addition to standard conditioning regimens for HCT, high-dose radioimmunotherapy (RIT) offers the unique opportunity to selectively deliver a high dose of radiation to the bone marrow while limiting side effects. Modification of a CD66b-specific monoclonal antibody (mAb) with a DTPA-based chelating agent should improve the absorbed dose distribution during therapy. The stability and radioimmunoreactive fraction of the radiolabeled mAbs were determined. Before RIT, all patients underwent dosimetry to determine absorbed doses to bone marrow, kidneys, liver, and spleen. Scans were performed twenty-four hours after therapy for quality control. A radiochemical purity of >95% and acceptable radioimmunoreactivity was achieved. Absorbed organ doses for the liver and kidney were consequently improved compared to reported historical data. All patients tolerated RIT well with no treatment-related acute adverse events. Complete remission could be observed in 4/5 of the patients 3 months after RIT. Two patients developed delayed liver failure unrelated to the radioimmunotherapy. The improved conjugation and radiolabeling procedure resulted in excellent stability, radiochemical purity, and CD66-specific radioimmunoreactivity of ^90^Y-labeled anti-CD66 mAb. RIT followed by conditioning and HCT was well tolerated. Based on these promising initial data, further prospective studies of [^90^Y]Y-DTPA-Bn-CHX-A″-anti-CD66-mAb-assisted conditioning in HCT are warranted.

## 1. Introduction

For patients with high-risk acute myeloid leukemia (AML), acute lymphoblastic leukemia (ALL), or myelodysplastic syndrome (MDS), allogeneic hematopoietic cell transplantation (HCT) is so far the only potentially curative treatment [1]. Traditionally, allogeneic HCT is preceded by a preparation regimen, the conditioning, which consists of high-dose chemotherapy and, in selected cases, external beam radiation therapy (EBRT). The conditioning regimen and the graft versus leukemia effect exerted by the donor cells play an essential role in determining disease control after transplant. Since disease relapse is one of the most frequent causes of treatment failure and death after allogeneic HCT, there is still an urgent need to implement the anti-leukemic power of such treatment without increasing its toxicity profile. In this context, high-dose radioimmunotherapy (RIT) offers the unique opportunity to selectively deliver a high radiation dose to the bone marrow with limited side effects, e.g., by targeting the CD66 antigen family that is highly expressed in the myeloid lineage of normal hematopoiesis [2,3]. This strategy can be used to either intensify standard conditioning or augment reduced-intensity conditioning to improve the outcome of HCT. The specificity and therapeutic efficacy of radiolabeled monoclonal antibodies (mAb), in this work antibodies directed against cells expressing CD66b (NCA-95; CEACAM-8) antigens [4,5,6,7], are highly dependent on the choice and coupling method of the chelator, the technique of radiolabeling, and the appropriate selection of the radionuclide. While in recent years data on ^188^Re-labeled anti-CD66-mAb (also known as besilesomab) for intensified conditioning before HCT have been reported, ^188^Re has disadvantages due to increased probability of nephrotoxicity as ^188^Re dissociates from the mAb [2,8,9,10,11]. Yttrium-90 (^90^Y) is a good alternative radionuclide in this respect; promising results have been achieved with ^90^Y-labeled anti-CD66-mAb to intensify of conditioning before HCT [10,12]. After our group’s initial studies, there are two recent abstracts only [13,14] on ^90^Y-radiolabeling of anti-CD66-mAb. The previously reported method of ^90^Y-radiolabeling after conjugation of the mAb with the chelator mx-DTPA (1-(4-isothiocyanatobenzyl)-4-methyldiethylenetriaminepentaacetic acid) was found to have poor reproducibility with a radiochemical purity of <90%, thereby theoretically increasing the risk of ^90^Y-uptake in critical organs like the kidneys or the liver. However, while the kidney dose was decreased compared to ^188^Re-labeled anti-CD66-mAb, there was an increased liver uptake of the previous ^90^Y-labeled anti-CD66-mAb observed by Ringhoffer et al. [12]. In addition to the potential for improvement regarding off-target accumulation, the antibody is now available in a reduced form as a Scintimun kit, so an adaptation of the preparation protocol will enable a broader application in the future. According to our research, the previously used chelating agent mx-DTPA is also challenging to obtain, so an adaptation was also recommended in terms of an easily applicable protocol. For conjugates using the chelating agent *p*-SCN-CHX-A″-DTPA as an alternative, superior stability in man was demonstrated compared to the originally used mx-DTPA [15,16].

Here we report the methodology of this procedure for ^90^Y-labeling of a *p*-SCN-CHX-A″-DTPA-conjugated anti-CD66-mAb (DTPA-Bn-CHX-A″-anti-CD66) for radioimmunotherapy, complemented with its first clinical data.

## 2. Materials and Methods

For these treatments, the antibody was labeled with the radionuclides indium-111 (^111^In) or technetium-99m (^99m^Tc) for dosimetry measurements. For therapy, yttrium-90 (^90^Y) was used for labeling. The different methods are described below.

### 2.1. Conjugation of Anti-CD66-mAb with p-SCN-CHX-A″-DTPA

The clonal antibody BW 250/183 (besilesomab) was obtained from Curium/Cisbio International (Scintimun, Gif sur Yvette Cedex, France). Before the conjugation of the mAb with the chelator DTPA (via *p*-SCN-Bn-CHX-A″-DTPA, Macrocyclics, Dallas, TX, USA), 16 mg of the mAb was dissolved in 1600 µL of 0.05 M HEPES-buffer (4-(2-hydroxyethyl)-1-piperazineethanesulfonic acid; pH 7.9). By incubation at room temperature for 72 h, the sulfide bonds of the reduced mAb were gently recombined. The successful recombination was confirmed using size exclusion chromatography (SEC; NGC Scout 10 Plus, Biorad, Hercules, CA, USA). Thereafter, 38.2 mg *p*-SCN-Bn-CHX-A″-DTPA was dissolved in 1600 µL of 0.05 M HEPES buffer, and the pH was adjusted to 8.5 with 6% NaOH solution. A total volume of 2 mL was achieved by adding 0.05 M HEPES buffer. Both the solutions were mixed with equal volumes of 1600 µL each, allowed to react for 2 h at ambient temperature, and subsequently stored for 15 h at 4–8 °C. Size exclusion chromatography (SEC; NGC Scout 10 Plus, Bio-Rad, Hercules, CA, USA) with phosphate-buffered saline (PBS)/HCl-buffer (pH 6.75) elution buffer was used for mAb purification and concentration (column: Superdex^TM^ 200 pg, GE Healthcare, Solingen, Germany). Fractions of the mAb were collected and pooled. Subsequently, the DTPA-Bn-CHX-A″-anti-CD66 containing fraction was washed two times with 0.9% NaCl using ultrafiltration via centrifugation for 15 min at 3000 rpm at room temperature (centrifuge: Heraeus Multifuge 3 S-R (Hanau, Germany) with Amicon Ultra 15 Centrifugal Filters (30 kDa, Merck Millipore, Tullagreen, Carrigtwohill, Ireland)). The concentrated antibody solution was then analyzed with the spectrophotometer Multiskan Go (Thermo Scientific, Waltham, MA, USA), using a µDrop plate and a predesigned protocol from Thermo Scientific at wavelengths of 260, 280, and 320 nm to determine the antibody concentration. Afterwards, aliquots of 500 µg of the antibody solution were prepared for the labeling with ^111^In (*n* = 1), or aliquots of 1500 µg of the antibody solution were prepared for the labeling with ^90^Y (*n* = 5). The aliquots were stored at −20 °C for subsequent radiopharmaceutical productions.

The chelator-to-mAb ratio was determined in a slightly modified way, as described before [17]. Briefly, the principle is based on a colorimetric reaction between an yttrium-arsenazo III complex (Y(AAIII)_2_) and diethylenetriaminepentaacetate (DTPA). Instead of using 3 mL cuvettes, the test was adapted to 96-well plates, and the absorbance was measured at 652 nm with a spectrophotometer (Multiskan Go, Thermo-Fisher, Waltham, MA, USA).

### 2.2. Labeling, Purification, and Serum Stability of [^111^In]In-DTPA-Bn-CHX-A″-Anti-CD66

185 MBq of ^111^In were obtained from Curium (Mallinckrodt medical B. V., Le Petten, the Netherlands) in 500 µL 0.02 molar HCl-solution. The pH of the ^111^In-solution was adjusted with 0.1 M NaOAc-solution (pH 7.0) to a value between 5 and 6. Thereafter, 500 µg of the DTPA-Bn-CHX-A″-anti-CD66 containing solution in 0.9% NaCl was added. The reaction mixture was kept at room temperature for 1 h. Afterwards, an aliquot of the reaction solution was analyzed using SEC-radio-HPLC (Pump UltiMate 3000 (Thermo-Fisher, Waltham, MA, USA), BioSep SEC-s3000 column, UV-detector (UVD340U, Gynkotek/Thermo-Fisher, Waltham, MA, USA), Gabi gamma-HPLC-detector (Elysia-Raytest, Straubenhardt, Germany)).

The purification of [^111^In]In-DTPA-Bn-CHX-A″-anti-CD66-mAb was performed using ultrafiltration with a Heraeus Multifuge 3 S-R (Hanau, Germany) and Amicon Ultra 15 Centrifugal Filters (30 kDa, Merck Millipore, Tullagreen, Carrigtwohill, Ireland) for 15 min at 3000 rpm.

To investigate serum stability, 150 µL of the radiolabeled mAb was added to 500 µL of human serum. The resulting mixture was incubated at 37 °C, and aliquots for radio-HPLC analysis were taken at 24, 48, and 72 h. The samples were analyzed using the HPLC conditions mentioned before.

### 2.3. Labeling, Purification, and Serum Stability of [^90^Y]Y-DTPA-Bn-CHX-A″-Anti-CD66

For radiolabeling, 8600 MBq of ^90^Y was obtained from Eckert & Ziegler (Berlin, Germany) in a 0.04 molar HCl solution. The pH of the ^90^Y solution was adjusted with 0.1 M NaOAc-buffer (pH 7.0) to a value between 5 and 6. Then a solution of 0.4 M NaOAc with 2.5% gentisic acid (pH 4.0) was added, resulting in a pH of 5–6. A 1500 µg DTPA-Bn-CHX-A″-anti-CD66 solution was added and left at room temperature for 10 min. Afterwards, an aliquot of the reaction solution was analyzed using radio-HPLC (Pump UltiMate 3000 (Thermo-Fisher, Waltham, MA, USA), BioSep SEC-s3000 column, UV-detector (UVD340U, Thermo-Fisher, Waltham, MA, USA), Gabi gamma-HPLC-detector (Elysia-raytest, Straubenhardt, Germany)), and the reaction mixture was purified with a SEC-system (SEC, NGC Scout 10 Plus, Biorad, Hercules, CA, USA; column: Yarra s3000 SEC-column, Phenomenex, Aschaffenburg, Germany) with an external γ-HPLC-detector (Gabi, Elysia-raytest, Straubenhardt, Germany) using ChromLab software ver. 6.0.0.34. As eluent for the purification PBS with gentisic acid (625 mg in 1000 mL PBS; pH 6.6–6.8 adjusted with 30% NaOH) and a flow rate of 4 mL/min was used. The product-containing fraction was eluted from the column after ~13 min. It was collected for the next 5 min, resulting in a total amount of 3991 ± 973 MBq (50–57% radiochemical yield, not decay corrected) ^90^Y-labeled DTPA-Bn-CHX-A″-anti-CD66 ([^90^Y]Y-DTPA-Bn-CHX-A″-anti-CD66-mAb).

Serum stability was assessed at 0.5 h and 24 h, as described for ^111^In-labeled anti-CD66 mAb.

### 2.4. Determination of Radiochemical Purity of [^90^Y]Y-DTPA-Bn-CHX-A″-Anti-CD66-mAb

For determination of the radiochemical purity of the [^90^Y]Y-DTPA-Bn-CHX-A″-anti-CD66-mAb, a system comprising an HPLC-pump (UltiMate 3000, Thermo-Fisher, Waltham, MA, USA), a BioSep SEC-s3000 column, a UV-detector (UVD340U, Thermo-Fisher, Waltham, MA, USA), a Gabi gamma-HPLC-detector (Elysia-raytest, Straubenhardt, Germany) and a PC with Chromeleon (ver. 7.2.9) was used. A PBS solution with a pH between 6.6 and 6.8 (adjusted with hydrochloric acid (30%), Honeywell) was used as the mobile phase with a flow rate of 1 mL/min. Additionally, the radiochemical purity was monitored using thin-layer chromatography (TLC) using RP-18 aluminum plates (silica-gel RP-18 F254, Merck-Millipore, Darmstadt, Germany) as stationary phase and 0.1 M ammonium acetate (pH = 5.7) with EDTA (50 mM) as mobile phase. It is known that the mAb remains at the starting point (R_f_ = 0), whereas ^90^Y moves with the mobile phase front (R_f_ = 1). For evaluation of the TLC plates, an FLA-3000 phosphorimager (Fuji Photo Film Co., Ltd., Tokyo, Japan) in combination with the control software BASReader (version 3.14, Elysia-raytest, Straubenhardt, Germany) and the image analyzer software AIDA (version 4.24.036, Elysia-raytest, Straubenhardt, Germany) was used.

### 2.5. Determination of Radioimmunoreactivity (RIR)

Granulocytes were isolated from leukocyte–platelet concentrate using Percoll^®^ (Miltenyi Biotec, Bergisch Gladbach, Germany) according to the manufacturer’s protocol. Cell numbers in the range of 10^4^–10^8^ were distributed on Eppendorf reaction vessels, and 1 ng radiolabeled mAb was added per batch. The batches (triplicates) were incubated in an overhead shaker for 60 min. After separating the unbound mAb, the absorbed activity was determined using an automated gamma-counter COBRA II (Perkin Elmer, Waltham, MA, USA). A general theoretical model was adapted to the experiment, and the measured data were fitted using GraphPad Prism ver. 9.5.1 (RRID:SCR_002798) [18].

### 2.6. Dosimetry

A detailed description of the method for performing dosimetry is beyond the scope of this manuscript. Briefly, absorbed doses were calculated based on pre-therapeutic planar imaging using [^99m^Tc]Tc-anti-CD66-mAb (commercially available kit, Scintimun, Curium) and a physiologically-based pharmacokinetic model [19,20,21]. The image acquisitions were performed on a Symbia T2 (Siemens Medical Solutions, Malvern, PA, USA; VB50B, Syngo MI App 2009A ver. 8.1.15.7) equipped with low-energy high-resolution parallel-hole collimators. Measurements were performed with an energy window at 140 keV (width 15%), and the dual-energy method was used for scatter correction (width of lower energy window 15%).

For the first patient, ^111^In was used for radiolabeling anti-CD66-mAb for dosimetry. After that, for logistical reasons mainly related to the unavailability of ^111^In, dosimetry was performed in the other 4 patients using [^99m^Tc]Tc-anti-CD66-mAb. The anti-CD66 antibody was labeled with technetium-99m (^99m^Tc) according to the instructions in the Scintimun Kit (Curium, London, UK).

### 2.7. Radioimmunotherapy

The treatment steps, including the method of patient selection, the time interval between dosimetry and RIT, and the time interval between RIT and conditioning, as well as the follow-up protocol, remain essentially unchanged compared with previously published data from our group [12]. Upon successful radiolabeling, the RIT program of Ulm University was reinitiated. Data from five patients in whom an interdisciplinary tumor board had decided to perform RIT were analyzed. Patients with high-risk acute leukemia and in complete remission or with high-risk MDS were selected. Other criteria were the availability of a matched donor and adequate organ function. Prior to RIT, all patients underwent dosimetry to determine the absorbed dose to red bone marrow and critical organs. All patients were referred by their treating immunology and stem cell transplantation physician for anti-CD66-radioimmunotherapy under compassionate use in compliance with the German Medicinal Products Act, AMG §13(2b). Patients gave written informed consent for RIT according to German law. Treatment and data analyses were performed in accordance with the 1964 Declaration of Helsinki and its subsequent amendments or comparable ethical standards and in accordance with Good Clinical Practice (GCP). The radiopharmaceuticals were produced in accordance with the German Medicinal Products Act §13(2b) and the relevant regulatory authorities.

No human anti-mouse antibodies (HAMA) test was performed before RIT; however, as recommended for RIT, all premedications were kept ready for urgent use during the procedure. To avoid radiolysis of mAb, total radioactivity was diluted in 5 mL and injected within 60 min of radiolabeling. RIT was performed as a slow infusion over 2 min in all patients over an indwelling central venous catheter. Vital signs of the patients were continuously monitored during and until 1 h after therapy for any side effects. Post-therapy scans (anterior and posterior views) were performed 1 d p.i. using a Symbia T2 camera (Siemens Medical Solutions, Malvern, PA, USA; VB50B, Syngo MI App 2009A ver. 8.1.15.7) equipped with medium-energy collimators. Measurements were performed with an energy window between 122 and 226 keV.

## 3. Results

### 3.1. Conjugation of the Chelator

Conjugation of the mAb with *p*-SCN-CHX-A″-DTPA (Figure 1A) followed by purification using SEC and subsequent ultrafiltration led to a pure and stable DTPA-Bn-CHX-A″-anti-CD66-mAb within two days. The purification process resulted in a loss of 31% of anti-CD66-mAb. Finally, 11 mg of conjugated DTPA-Bn-CHX-A″-anti-CD66 was obtained in a concentration of 3 to 4 mg/mL for radiolabeling. The determined chelator-to-mAb ratio was 4.6 ± 0.3 (*n* = 3; range 4.3–5.1).

### 3.2. Radiolabeling, Purification, and Stability of [^111^In]In-DTPA-Bn-CHX-A″-Anti-CD66

The radiolabeling procedure for ^111^In-labeled and ^90^Y-labeled antibodies was almost identical (Figure 1B). A longer reaction time was needed for ^111^In, and the addition of gentisic acid was omitted as no radiolysis effect was expected due to the small amount of ^111^In activity and the low ionizing energy of the radionuclide.

The stability of the DTPA-Bn-CHX-A″-anti-CD66-mAb was tested in human serum for three days with the ^111^In-labeled DTPA-Bn-CHX-A″-anti-CD66-mAb, demonstrating that after 72 h up to 82% ± 2% (*n* = 3) of the mAb were still intact.

### 3.3. Radiolabeling, Purification, and Stability of [^90^Y]Y-DTPA-Bn-CHX-A″-Anti-CD66

After radiolabeling and subsequent quality control using SEC, 85–96% (mean 93% ± 4%) pure ^90^Y-labeled mAb was obtained (Figure 1B). Subsequent purification with a SEC-system raised the purity of the radiolabeled mAb to 96–99% (mean 98% ± 1%; Figure 2).

Post-purification yields were determined to be 50–57% (average 53% ± 3%), corresponding to a mean of (4630 ± 273) MBq (not decay corrected). For quality control assays, aliquots were collected from the whole batch. A mean of (4343 ± 182) MBq (not decay corrected) of ^90^Y-labeled DTPA-Bn-CHX-A″-anti-CD66 was mixed with 0.625 mg/mL gentisic acid in ~20 mL PBS buffer (pH = 6.6–6.8). Radiochemical purity tests in the aliquots revealed a loss of less than 5% of intact ^90^Y-labeled mAb up to 3 h after the radiolabeling and purification. The retention time for [^90^Y]Y-DTPA-Bn-CHX-A″-anti-CD66-mAb was found to be 8 min, whereas, for unbound ^90^Y, a retention time of 12 min was detected. Another peak at ~7 min was observed, which can be assigned to the dimeric mAb. This dimerization of mAb cannot be prevented as the mAb is highly concentrated after the conjugation with *p*-SCN-CHX-A″-DTPA. Radio-TLC measurements revealed radiochemical purities of 92–96%. Purification using ultrafiltration instead of SEC resulted in a radiochemical purity of 88% ± 2% with a 68% ± 5% yield.

Similar results with a mean purity of 94% ± 1% using TLC confirmed the quality control of SEC. The measured radiochemical purities for the purified mAb were, on average, 3% lower than the results of the HPLC measurements. We speculate that a small amount of ^90^Y is retained on the HPLC column and therefore remains undetected. Also, radiolytic effects potentially play a role, as measurements of the TLC plates were performed 1–2 h after the HPLC measurement.

Radioimmunoreactivity was 0.66 ± 0.03 for a specific activity of 5.73 MBq/µg [^90^Y]Y-DTPA-Bn-CHX-A″-anti-CD66.

The stability of ^90^Y-labeled antibodies was slightly lower with (70 ± 2)% after 24 h (*n* = 4) compared to ^111^In-labeled mAb, probably due to radiolysis effects (Appendix A, Appendix A).

### 3.4. Dosimetry and RIT

Five patients (MDS, 2; c-ALL, 1; AML, 2) were administered (3.63 ± 1.05) GBq (1.79 GBq–4.23 GBq) of [^90^Y]Y-DTPA-Bn-CHX-A″-anti-CD66-mAb. Table 1 summarizes the clinical data of the patients, including prior therapies and conditioning regimes. Dosimetry was successfully performed with both ^111^In-labeled and ^99m^Tc-labeled mAb. Based on dosimetry and post-therapy scintigraphy, all five patients achieved median (range) doses of 24.3 (9.4–26.3), 6.0 (3.5–7.0), and 2.1 (0.9–2.5) Gy in bone marrow, liver, and kidneys, respectively (Table 2).

Post-therapy scintigraphy performed 1 day post RIT revealed negligible visual uptake in kidneys and intestines. As expected and analogous to [^99m^Tc]Tc-anti-CD66-mAb scintigraphy, the post-therapy scintigraphy confirmed high uptake in the bone marrow and spleen; in the liver, mild to moderate tracer retention was observed. The obtained absorbed doses and post-therapy scintigraphy images are presented in Figure 3.

The absorbed doses are compared to previously reported data in Figure 4.

Post-RIT laboratory values showed absolute neutrophil counts declining to a median value of 0.4 (range 0.1–1) Giga/L (normal range 1.3–6.7) 7 days after RIT, dropping to 0 Giga/L on day 12 after RIT (Appendix A). All patients tolerated RIT well without any therapy-associated acute side effects. After conditioning (7–8 days post RIT), two patients developed liver failure, one of which died. Follow-up until 3 months post RIT revealed complete remission (CR) in 4/5 patients.

The patient (number 3 in Table 2) who developed liver failure and died due to veno-occlusive disease (VOD) was treated with inotuzumab–ozogamicin and received his second HCT, thus presenting a high risk for developing of a VOD. The liver biopsy of patient 4 (Table 2), who developed liver failure symptoms 4 months after RIT, showed no signs of cirrhosis. A VOD and a transplant-associated microangiopathy were also excluded. Relevant hemosiderosis was detected. In Appendix A, the time course of transaminases, GGT, and total bilirubin after RIT of patient 3 and patient 4 is depicted.

## 4. Discussion

Conjugation and radiolabeling of an anti-CD66-mAb with ^90^Y and ^111^In using *p*-SCN-Bn-CHX-A″-DTPA as a chelating agent was successfully carried out, and a more favorable off-target accumulation was obtained compared to the previous modified version of the antibody. Radioimmunotherapy in the first five patients for intensification of conditioning before allogeneic HCT resulted in high doses in the bone marrow with only limited doses in critical organs.

Conjugation of the anti-CD66-mAb with *p*-SCN-CHX-A″-DTPA, purification of the conjugated mAb, and subsequent radiolabeling with ^111^In or ^90^Y were successfully performed. Also, after radiolabeling and subsequent purification, good yields with high radiochemical purity and good labeling stability in serum could be achieved.

For labeling with ^188^Re [9,22] or ^99m^Tc, the disulfide bonds of the hinge region of the antibody need to be reduced. The respective commercially available antiCD66-kit-antibody from Scintimun is already provided in reduced form. For chelator conjugation and subsequent labeling with ^90^Y or ^111^In, an antibody in a non-reduced form is necessary.

While the previous ^90^Y-labeling protocol was based on a non-reduced antibody, the anti-CD66 mAb is nowadays only commercially available in a reduced form. Therefore, the disulfide bonds need to be recovered before the conjugation of the chelator for ^90^Y-labeling, demanding an additional step in the conjugation protocol. Incubation of the antibody under mild conditions in 0.05 M HEPES buffer (pH 7.9) for 3 days could reproducibly restore the disulfide bonds and recover the mAb in a form indistinguishable from the native anti-CD66-mAb using SEC. However, it is not excluded that a good antibody recovery can also be achieved with shorter incubation times. Various studies on antibody reduction and reoxidation show a wide range of available methods and incubation times [23,24,25,26,27,28].

Conjugation with the new chelator increased labeling stability and improved non-specific accumulation compared to the previous ^90^Y-labeled mAb. While several chelators were tested for labeling with ^90^Y [29,30,31], DTPA is advantageous as it allows for radiolabeling at 37 °C and below. Therefore, it is an ideal chelating agent for temperature-sensitive compounds like mAbs [31]. The previously used mx-DTPA (tiuxetan) is a well-known DTPA derivative often used for labeling with ^90^Y, especially in radioimmunotherapies [32,33,34,35,36,37]. This chelating agent is no longer readily available, and switching to a broadly accessible ligand is certainly advantageous for the broad clinical acceptance of the therapy. The chelating agent *p*-SCN-CHX-A″-DTPA ([(*R*)-2-amino-3-(4-isothiocyanatophenyl)propyl]-trans-(*S*,*S*)-cyclohexane-1,2-diamine-pentaacetic acid) is very promising for conjugation and subsequent labeling of antibodies with ^90^Y. It is applied for labeling diabodies [38] and various antibodies [39,40,41,42,43,44,45]. Furthermore, it has the advantage of commercial availability compared to mx-DTPA and higher stability compared to other DTPA derivatives has been demonstrated [29,46].

The use of SEC for purification guaranteed highly pure radiolabeled mAb. The purity after using ultrafiltration was on average 5% lower. However, purification via SEC also leads to a higher product loss than ultrafiltration. This might be caused by adhesive effects of the mAb on surfaces, like glass (e.g., the reaction vial), synthetic materials (e.g., syringes), or the filling material of the purification column. A potential solution to reduce the high loss of ^90^Y-labeled anti-CD66-mAb and in consequence to reduce the starting activity of ^90^Y is the use of smaller purification columns, with less filling material on which the mAb can be retained. Also, using a conditioning substance that inhibits the mAb from binding to surfaces (e.g., human serum albumin) might be beneficial. Moreover, the first efforts to improve the ultrafiltration method provided promising data with a higher yield at comparable purity.

A good serum stability of the [^90^Y]Y-DTPA-Bn-CHX-A″-anti-CD66-mAb was observed. As it is known that antioxidants can be used to prevent the fragmentation of organic molecules [47], we used gentisic acid as a scavenger during radiolabeling, purification, and storage of the mAb for up to 3 h before patient treatment.

As for radiochemical purity, a small shoulder in the SEC profile before the main peak is expected due to a proportion of dimeric mAb. This is most likely a consequence of conditions during the storage and handling of the mAb and is a phenomenon known from other antibodies as well [48,49,50]. Therefore, dimerization was tolerated, as was a small proportion of unbound ^90^Y and mAb fragments, and they had no noticeable effect on the efficacy of RIT.

For radioimmunoreactivity, the values determined were reasonable and to be expected due to the modification of the antibody. Radiolytic effects might have an additional influence on the radioimmunoreactivity for the test, as mAb concentration was higher compared to the patient application. Optimization of the test method regarding antibody concentration and activity concentration is ongoing.

The clinical results of RIT using [^90^Y]Y-DTPA-Bn-CHX-A″-anti-CD66-mAb for intensification of conditioning before allogeneic HCT in the first patients were very promising. RIT with [^90^Y]Y-DTPA-Bn-CHX-A″-anti-CD66-mAb resulted in high doses in the bone marrow of up to 26 Gy in all patients with only moderate doses in the liver and kidneys, which means excellent target-to-critical-organ ratios. The absorbed doses in the kidneys and the liver of CHX-A″-DTPA-conjugated and ^90^Y-labeled anti-CD66-mAb presented here are substantially lower compared to the previous version of the ^90^Y-labeled mAb (Figure 4), which we believe is an excellent improvement for further therapeutic application.

Also, the accumulation in the spleen did not change compared to the previous ^90^Y-labeled and the ^188^Re-labeled antibodies. The hematogenous red pulp of the spleen is known to contain granulocytes, which are the target of anti-CD66-mAb [51,52,53,54,55,56]. The accumulation of the antibody in the bone marrow and spleen depends on the amount of maturing granulopoiesis in these organs and is likely to vary considerably. The low BM dose in patient 1 is probably due to the diagnosis of hypocellular MDS; the low spleen dose in patient 5 is unexplained.

When comparing these data to our historical data and data from the literature on RIT for intensification of conditioning before HCT, the target-to-critical-organ ratios are comparable or even superior.

All five patients tolerated RIT without any immediate acute side effects. After the conditioning regimen and the subsequent transplantation, two patients developed symptoms of liver failure. Compared with the data of the ^188^Re-labeled antibody as well as the previous ^90^Y-labeled variant, the liver uptake could be equalized or significantly reduced. However, according to the clinical data and patients’ histories, this was probably unrelated to RIT: one patient who finally died due to liver failure was receiving his second transplant and had a high pre-HCT probability of developing VOD due to prior treatment with inotuzumab–ozogamicin and second transplantation, as previously shown [57]. The reason for the development of liver failure in patient 4 is unclear. The most frequent causes of liver failure after HCT, including viral infections, graft versus host disease, VOD, and disease reoccurrence, were ruled out. As an exclusion diagnosis, the hypothesis of idiopathic ascites was considered [58]. Apart from these most likely not RIT-associated adverse events, no other adverse events were observed during follow-up. Clinical and laboratory values showed CR in 4/5 patients after 3 months.

## 5. Conclusions

Our improved conjugation and radiolabeling procedure using *p*-SCN-Bn-CHX-A″-DTPA resulted in high stability, purity, and radioimmunoreactivity of [^90^Y]Y-DTPA-Bn-CHX-A″-anti-CD66-mAb. The modified version of the mAb showed superior characteristics to its predecessor when comparing the doses absorbed by the kidneys and liver. Due to the promising first clinical results, RIT with [^90^Y]Y-DTPA-Bn-CHX-A″-anti-CD66-mAb for augmenting conditioning before allogeneic HCT has been reintroduced in the clinical routine at our institution. Based on these promising preliminary data, future large-scale prospective trials are justified and warranted to evaluate the clinical value and potential of RIT in this context.

## Figures and Tables

**Figure 1 cancers-15-03660-f001:**
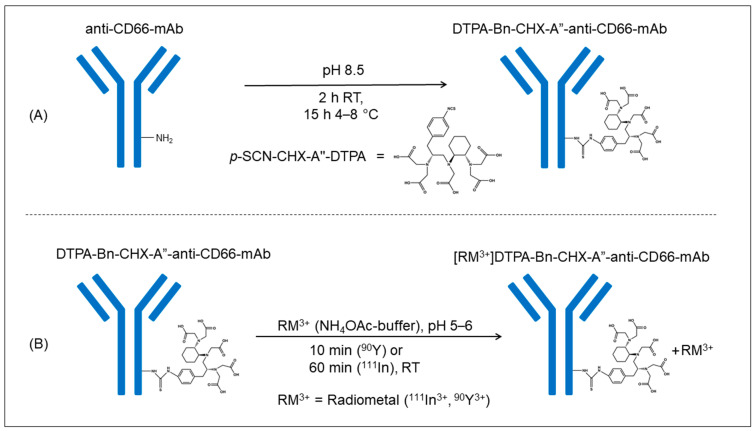
Graphical visualization of the conjugation-reaction process of the anti-CD66-mAb with the chelator p-SCN-CHX-A″-DTPA (**A**) and the subsequent radiolabeling procedure of the DTPA-anti-CD66-mAb with the radiometal (^90^Y, ^111^In, ^99m^Tc) (**B**).

**Figure 2 cancers-15-03660-f002:**
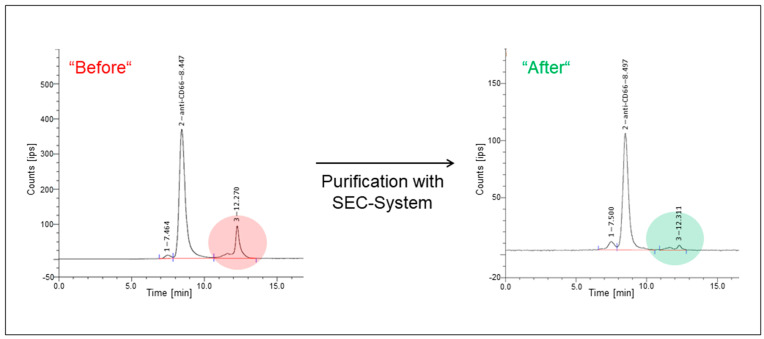
Exemplary radiochromatograms of the ^90^Y-labeled anti-CD66-mAb before and after SEC-purification. Radiolabeled DTPA-CHX-A″-anti-CD66-mAb (anti-CD66) was detected at a retention time of ~8.5 min. A small fraction of radiolabeled mAb-complexes was observed at 7.5 min, while unbound ^90^Y was seen at a retention time of 12.3 min. The unbound ^90^Y fraction was reduced by SEC purification (color highlighted areas).

**Figure 3 cancers-15-03660-f003:**
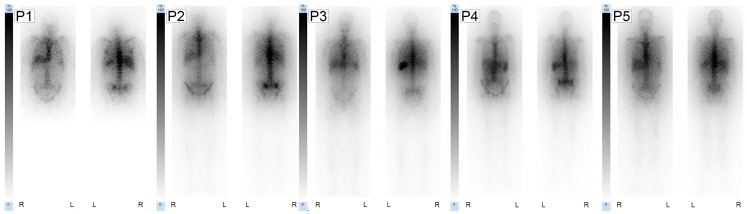
Post-therapy anterior and posterior images after RIT are shown for each patient (P1–P5). Whole-body planar scintigraphies were obtained 1 d after RIT started on a Siemens Symbia T2 using the ME collimator. For each patient image the scale was normalized to the individual liver value (100%).

**Figure 4 cancers-15-03660-f004:**
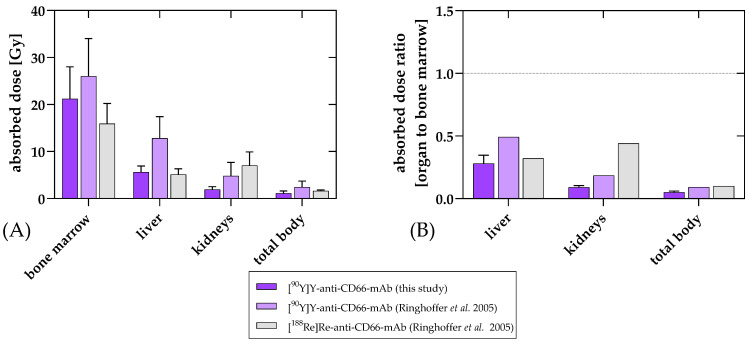
Organ-absorbed doses (mean and standard deviation) (**A**) and ratios of the organ-absorbed doses to bone marrow dose (**B**) of [^90^Y]Y-anti-CD66-mAb for the current treatments compared to the data published by Ringhoffer et al. [12]. Absorbed doses were determined and compared for bone marrow, kidneys, liver, and remainder. While uptake in the bone marrow as a target region was quite similar to the previous ^90^Y-labeled mAb, non-specific uptake in liver and kidney was favorably reduced. This was also reflected in the results for total body. The preferred distribution was also suggested when comparing the ratio of organs to bone marrow.

**Table 1 cancers-15-03660-t001:** Clinical data of the patients.

Patients’ Characteristics	
Number	5
Age at the time of HCT	
Median	63
Range	48–65
Sex (m:f)	4:1
Diagnosis (n, %)	
AML	2 (40%)
MDS	2 (40%)
ALL	1 (20%)
Disease stage at HCT (n, %)	
Upfront/early	2 (40%)
CR	3 (60%)
Donor type (n, %)	
MUD	5/5 (100%)
Conditioning regimen (n, %)	
Fludarabin, treosulfan + ATG	2 (40%)
Fludarabin, melphalan, carmustin + ATG	2 (40%)
Fludarabin, thiotepa, carmustin + ATG	1 (20%)
GvHD prophylaxis (n, %)	
Tacrolimus + MMF	5/5 (100%)
Number of HCT (n, %)	
1st HCT	3/5 (60%)
2nd HCT	2/5 (40%)

**Table 2 cancers-15-03660-t002:** Absorbed doses of the patients for bone marrow, liver, spleen, kidneys, and whole body. EBRT = external beam radiation therapy.

Organ Dose [Gy]	P1 ^a^	P2	P3	P4	P5
Bone marrow	9.4	21.7	24.5	26.3	24.3
Liver	3.5	7.0	6.3	5.3	6.0
Spleen	10.8	16.1	33.4	23.1	42.2
Kidneys	0.9	2.4	2.2	2.1	1.8
Whole body	0.5	1.1	1.0	1.4	1.7

^a^ Patient received additional EBRT.

## Data Availability

The used data, in addition to those in the supplementary materials, are available from the corresponding author upon reasonable request.

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
