# Peer review of "Optimization of Radiolabeling of a [90Y]Y-Anti-CD66-Antibody for Radioimmunotherapy before Allogeneic Hematopoietic Cell Transplantation"

_cancers, 2023, doi:10.3390/cancers15143660_

Round 1

Reviewer 1 Report

The authors described the results of phase 1 trial of 90Y-anti-CD66 antibody in patients who received stem cell transplant.

1. Please, add IRB number.

2. Two patients developed liver failure. Authors should mention the points more in detail.

Author Response

We thank the reviewer for the evaluation and answer the questions accordingly as follows:

  1. Please, add IRB number.

The presented data are a compilation of patient treatments performed according to national and international laws and regulations. A(German Law Gazette I, page 2192; date of approval: 19th of October 2012, substituted by: German Law Gazette I, page 3394; date of approval: 25th of June 2020). Therefore, for publishing a compilation of such legal treatments, no IRB number is necessary or obtainable.  

  1. Two patients developed liver failure. Authors should mention the points more in detail.

To clarify, we added, in addition to the results (lines 350, 353-359) and the discussion (lines 445-458), in the abstract (lines 43-44) the sentence “Two patients developed delayed liver failure unrelated to the radioimmunotherapy.“.

Reviewer 2 Report

Radioimmunotherapy is theoretically interesting but practically challenging. The authors did a nice job in radiolabeling a CD66 construct for the treatment of leukemia. The patient numbers are small and the liver toxicity in 2/5 patients is concerning, but overall, the data are well documented.

Author Response

Thank you for your supportive review.

Reviewer 3 Report

Relapse is the major problem following HCT for leukemia and MDS.Therefore this study is of major interest for HCT physicians.

This is an improvment of this groups longstanding experience using antl-CD 66 antibodies for conditioning in HCT recipeints.

The dosimetry,labeling,purification,radioimmunoreactivityand serumstability are all very well described.However, characteristics of the first five pilotpatients are missing.I would like to see a Table including,patients sex,age,diagnosis,disease stage,type of donor,donor sex,age, other conditioning,immunosuppressive therapy,previouse HCT etc.

Please discuss why Pt 1 had low doses in BM compared to Pt 2-4.Pt 5 had remarkable low doses in spleen.What are the reasons for such variability?

Author Response

Relapse is the major problem following HCT for leukemia and MDS. Therefore, this study is of major interest for HCT physicians.

This is an improvement of this groups longstanding experience using anti-CD66 antibodies for conditioning in HCT recipients.

The dosimetry, labeling, purification, radioimmunoreactivity and serum stability are all very well described. However, characteristics of the first five pilot patients are missing. I would like to see a Table including, patients sex, age, diagnosis, disease stage, type of donor, donor sex, age, other conditioning, immunosuppressive therapy, previous HCT etc.

As suggested, we added a table (new Table 1) presenting the desired information. Just donor age and sex were not added, as for such a small casuistic patient number, this information may lead to unjustified speculations.

Please discuss why Pt 1 had low doses in BM compared to Pt 2-4. Pt 5 had remarkable low doses in spleen. What are the reasons for such variability?

The following explanation is added in lines 487-491: „The accumulation of the antibody in the bone marrow and spleen depends on the amount of maturing granulopoiesis in these organs and is likely to vary considerably. The low BM dose in patient 1 is probably due to the diagnosis of hypocellular MDS; the low spleen dose in patient 5 is unexplained.“